# Consumption of Tritordeum Bread Reduces Immunogenic Gluten Intake without Altering the Gut Microbiota

**DOI:** 10.3390/foods11101439

**Published:** 2022-05-16

**Authors:** Carmen Haro, María H. Guzmán-López, Miriam Marín-Sanz, Susana Sánchez-León, Luis Vaquero, Jorge Pastor, Isabel Comino, Carolina Sousa, Santiago Vivas, Blanca B. Landa, Francisco Barro

**Affiliations:** 1Department of Crop Protection, Institute for Sustainable Agriculture—Spanish National Research Council (IAS—CSIC), 14004 Córdoba, Spain; charo@ias.csic.es (C.H.); blanca.landa@ias.csic.es (B.B.L.); 2Department of Plant Breeding, Institute for Sustainable Agriculture—Spanish National Research Council (IAS—CSIC), 14004 Córdoba, Spain; mmarin@ias.csic.es (M.M.-S.); ssanchez@ias.csic.es (S.S.-L.); 3Department of Gastroenterology, Hospital of León, Biomedicine Institute, University of León, 24071 León, Spain; luisvaqueroayala@gmail.com (L.V.); svivasa@gmail.com (S.V.); 4Novapan, S.L., C/Chopo, 68-70, 50171 La Puebla de Alfinden, Spain; jpastormoreno@gmail.com; 5Department of Microbiology and Parasitology, Pharmacy Faculty, University of Seville, 41004 Seville, Spain; icomino@us.es (I.C.); csoumar@us.es (C.S.)

**Keywords:** gliadin, gluten, glutenin, tritordeum, gluten-free diet, IgE binding, immunogenic gluten peptides, microbiota

## Abstract

Gluten proteins are responsible for the wheat breadmaking quality. However, gluten is also related to human pathologies for which the only treatment is a gluten-free diet (GFD). GFD has gained popularity among individuals who want to reduce their gluten intake. Tritordeum is a cereal species that originated after crossing durum wheat with wild barley and differs from bread wheat in its gluten composition. In this work, we have characterized the immunogenic epitopes of tritordeum bread and results from a four-phase study with healthy adults for preferences of bread and alterations in the gut microbiota after consuming wheat bread, gluten-free bread, and tritordeum bread are reported. Tritordeum presented fewer peptides related to gluten proteins, CD-epitopes, and IgE binding sites than bread wheat. Participants rated tritordeum bread higher than gluten-free bread. Gut microbiota analysis revealed that the adherence to a strict GFD involves some minor changes, especially altering the species producing short-chain fatty acids. However, the short-term consumption of tritordeum bread does not induce significant changes in the diversity or community composition of the intestinal microbiota in healthy individuals. Therefore, tritordeum bread could be an alternative for healthy individuals without wheat-related pathologies who want to reduce their gluten consumption without harming their gut health.

## 1. Introduction

Wheat is the most cultivated cereal in the world. This is mainly due to its unique viscoelastic properties. Wheat provides a quarter of the annual demand for plant proteins, carbohydrates, and dietary fiber [1]. Gluten proteins are responsible for providing wheat dough with its functional properties, allowing wheat to be the most widely used cereal for bread making. Gluten accounts for 80% of the total protein content of the grain and is classically categorized into two groups: primarily monomeric gliadins and polymeric glutenins [2]. Gliadins can be further divided into ω-, α/-, and γ-gliadins, while glutenins are comprised of the High Molecular Weight (HMW) and the Low Molecular Weight (LMW) glutenin subunits [3]. Based on genetic and proteomic research, a typical bread wheat cultivar may include 60 different gluten proteins in the grain, half of which are gliadins [4]. Other *Triticeae* species such as barley and rye, have gluten in their grain [5].

However, gluten proteins also trigger the immune response in celiac disease (CD) [6,7], which results in intestinal damage and malabsorption symptoms. CD is a chronic enteropathy associated with the human leukocyte antigen (HLA) HLA-DQ2 and HLA-DQ8 in susceptible people, which account for an increasing 1% of the population [8,9]. Even though several epitopes have been described in gluten peptides, the gliadin fraction contains the most stimulating epitopes in CD [10,11].

Gluten is also associated with non-celiac wheat sensitivity (NCWS), a pathology defined by extra-intestinal and intestinal symptoms after ingesting wheat-containing foods [12,13]. As opposed to CD, no autoimmune response or intestinal modifications are observed in NCWS. Apart from gluten, other proteins may be involved in this pathology, including the α-amylase/trypsin inhibitors (ATIs) and FODMAPs (fermentable oligosaccharides, disaccharides, monosaccharides, and polyols) [2,14,15]. Other pathologies related to gluten are gluten ataxia and allergies such as baker’s asthma and wheat-dependent exercise-induced anaphylaxis (WDEIA). Nowadays, the only effective treatment for wheat-related pathologies is to follow a life-long gluten-free diet (GFD).

Although GFD was intended primarily for wheat-related pathologies, it has been recognized as an optional treatment for other conditions such as dermatitis herpetiformis and irritable bowel syndrome (IBS) [16]. In fact, several studies with IBS patients often recommend the adherence to a GFD as it leads to a significant improvement in their symptoms: pain, bloating, stool consistency, or tiredness [15,17]. 

Moreover, this diet has recently gained popularity among non-diagnosed individuals who currently make up most adherents [16,18]. For instance, a 2018 Gallup poll stated that gluten-free products were actively incorporated into the diet by 21% of the USA population [19]. Additionally, Nielsen reported that 23% of the survey participants avoided gluten [20]. Nevertheless, a GFD presents certain drawbacks. In addition to being generally more expensive, gluten-free foods are less healthy than their gluten-containing counterparts as they are made with large amounts of fat and sugar to mimic the viscoelastic qualities of wheat bread [2]. Moreover, adherence to a GFD has proven to cause dysbiosis in gut microbiota, increasing bacterial populations considered unhealthy while reducing those considered beneficial [21]. This is associated with the modification of the quantity and quality of carbohydrates ingested in a GFD that leads to shifts of gut bacteria with the ability to degrade these carbohydrates to short-chain fatty acids (SCFA), which are known for their multiple benefits to human health [22,23,24,25].

Tritordeum (× *Tritordeum* Ascherson et Graebner) is a hexaploid cereal species produced by hybridization between two other cereals: wild barley (*Hordeum chilense*) and durum wheat (*Triticum durum*) [26]. It is characterized by its high carotenoid content; it presents a 4.8-fold increase compared to durum wheat [27]. Furthermore, as determined by ELISA R5, tritordeum contains about 49% less gluten content than wheat [13]. The same study in healthy individuals revealed that the amount of gluten immunogenic peptides found in the stools was significantly reduced when tritordeum bread was consumed instead of wheat bread. Although not suitable for celiac patients, tritordeum can be considered an alternative for NCWS patients who do not need strict gluten exclusion from their diet [28], and for people who wish to reduce their daily gluten intake. Moreover, the diet with Tritordeum-based foods (bread, bakery products, and pasta) significantly reduced IBS patients’ symptoms in a pilot study [29].

In line with this, we previously reported that the consumption of tritordeum bread, as well as a bread low in gliadins under a GFD, does not alter the structure and global composition of the intestinal microbiota in patients with NCWS. However, consumption of both breads increased the abundance of SCFA-producing bacteria, favoring a microbial profile that plays a key role in the intestinal health of these patients [5,28]. Nowadays, there is a tendency in the general population to move towards a GFD. However, few studies analyze the gut microbiota after the consumption of a GFD in healthy patients [21,30]; consequently, it would be of special interest to analyze the changes, if any, that may occur after this diet shift. 

Our study aims to delve into the characterization of the immunogenic epitopes of tritordeum bread further and compare them to wheat bread in addition to analyzing the gut microbiota after a GFD, as well as the effect of the consumption of tritordeum bread and wheat bread on healthy patients. This study may provide evidence for a healthy alternative to strict GFD for the population who wants to reduce gluten consumption in their diet but does not suffer any gluten-related pathology. To this, a four-phase study was designed to review preferences of bread and changes in the gut microbiota of healthy adults after consuming a gluten-containing bread of choice, gluten-free breads, a tritordeum bread and a given gluten-containing bread.

## 2. Materials and Methods

### 2.1. Design and Study Population

A study was designed to compare different breads with a prospective and within-subject study methodology. Twenty healthy adult subjects with negative serology for celiac disease and duodenal biopsy without alterations of the duodenal villi were selected. All had participated in a previous study to assess celiac disease in risk subjects and were first-degree adult relatives of celiac subjects [13]. All were on a regular diet with gluten at the time of the study and had no digestive diseases or symptoms. None of them presented other conditions or was under chronic medication. All study participants provided informed consent, and the Ethics Review Board approved the study design of the Hospital of León (approval number 1626).

### 2.2. Phases of the Study

The study comprised four phases, all lasting seven days (Figure 1). Briefly: (i) phase A: regular gluten diet. The patients incorporated the gluten-containing bread that they usually consume into their diet. No gluten restrictions were imposed to guarantee a minimum gluten intake of 100 g/day. (ii) Phase B: strict GFD. During this phase, patients consumed the gluten-free bread that their celiac relatives typically purchase. (iii) Phase C: continuation of the strict GFD. In this phase, a range of 100–150 g of Tritordeum bread was incorporated throughout their daily intake. (iv) Phase D: GFD for the last seven days of the study, in which a gluten-containing bread was also incorporated in the same format as Tritordeum bread (provided by the same company). Consumption was also a minimum of 100 and a maximum of 150 g/day, distributed over meals. The patients were blind to which type of bread was consumed during phases C and D. At the end of each phase, a stool sample (samples A, B, C, and D, respectively) and a clinical questionnaire testing the palatability and preference of bread were collected.

### 2.3. Preparation and Supplying of Bread Types

The preparation and supply of bread types were previously described by Vaquero et al. [13]. In brief, wheat breads were prepared using wheat sourdoughs, and after 24 h, breadmaking was performed. For that purpose, 120 g/kg wheat sourdough was mixed with wheat flour cv. ‘Bell’, water, salt, and yeast formed dough, which rested for 90 min at room temperature. For tritordeum breads, 150 g/kg of the tritordeum sourdough were used to create the dough, along with the rest of the common ingredients. The bulk doughs were divided into pieces and allowed to increase two-thirds of the mold volume before baking. After cooling, the loaves were cut into slices and frozen in portions.

The bread was stored frozen at the Hospital of León, where subjects received the bread for Phases C and D. Subjects were instructed to defrost the bread immediately before consumption. Subjects and staff were blinded to the type of bread provided in Phases C and D.

### 2.4. Protein Extraction and Digestion

White flour from wheat and tritordeum breads was subjected to protein extraction and digestion as described by Vaquero et al. [13]. Briefly, gluten protein extraction was performed using UPEX solution followed by 60% ethanol. After extraction, the total protein content of each sample was precipitated by the methanol/chloroform method, and concentration was determined. For digestion, 40 µg of protein pellets were resuspended, denatured, and reduced. Samples were diluted up to 120 µL to reduce guanidine concentration with 50 mM TEAB. Digestions were performed using sequence grade-modified trypsin (Sigma-Aldrich, Burlington, MA, USA) or chymotrypsin endoproteinase MS Grade (Thermo Scientific, Waltham, MA, USA), which were added to each sample in a 1/20 ratio (*w*/*w*) and then incubated at 37 °C overnight on a shaker. Sample digestions were evaporated to dryness.

### 2.5. Liquid Chromatography and Mass Spectrometer Analysis

Digested peptides of each sample were subjected to 1D-nano LC ESI-MS/MS analysis using a nano liquid chromatography system coupled to a high-speed Triple TOF 5600 mass spectrometer with a Nanospray III Source. The analytical column used was a silica-based reversed-phase column C18 ChromXP 75 µm × 15 cm, 3 µm particle size, and 120 Å pore size (Eksigent Technologies, AB SCIEX, Foster City, CA, USA). The trap column was a C18 ChromXP (Eksigent Technologies, AB SCIEX, Foster City, CA, USA), 3 µm particle diameter, 120 Å pore size, switched on-line with the analytical column. The loading pump delivered a 2 µL/min of 0.1% formic acid in water. The nano-pump provided a flowrate of 300 nL/min and was operated under gradient elution conditions for 150 min, using 0.1% formic acid in water and 0.1% formic acid in acetonitrile as mobile phase A and B, respectively.

### 2.6. Proteomic Data Analysis

Analyst^®^ TF 1.5.1 Software (AB SCIEX) was used to process MS and MS/MS data obtained for individual samples. The raw data files were converted to Mascot generic files (.mgf), then searched against *Triticum* spp. NCBI database using the Mascot Server v. 2.5.0 (Matrix Science, London, UK). The search parameters and additional details were described in Vaquero et al. [13].

Custom Python scripts were used to search canonical CD-epitopes, the p31–43 peptide, monoclonal antibodies recognition sites, and IgE binding sites in unique peptides with perfect match or one mismatch. These scripts were also used to sort these unique peptides into the different types of gluten proteins: ω-, α- and γ-gliadins, HMW, and LMW glutenin subunits. For these purposes, only peptides annotated as *Triticum aestivum*, *Triticum turgidum*, and *Hordeum* spp. proteins were considered for analysis. The sequences searched in the unique peptide database are described in Appendix A.

### 2.7. Collection and DNA Extraction of Stool Samples

At the end of each study phase, subjects were instructed to collect a 2–4 g stool sample into a sealed container after recording food intake. Specimens were delivered within the first two hours after deposition and were stored at −80 °C until processing. All samples were identified and labeled with a random numeric code.

The MoBio PowerSoil kit (QIAGEN, Venlo, The Netherlands) was used to extract the DNA from stool samples according to the manufacturer’s instructions, including a pre-step of high shaking, to improve lysis, using a TissueLyser II homogenizer (QIAGEN). DNA was quantified using a NanoDrop ND-1000 spectrophotometer (Nanodrop Technologies, Wilmington, DE, USA).

### 2.8. NGS and Bioinformatic Analysis of the Gut Microbiota

The V1–V2 hypervariable regions of the bacterial 16S rRNA of the 79 stool samples (10 patients per four phases of study and two DNA repetitions for each phase and patient, except one subject who has only one repeat of phase D) were amplified by PCR using the universal bacterial primers 8F (AGAGTTTGATCMTGGCTCAG) and 357R (CTGCTGCCTYCCGTA) and subjected to NGS analysis using an Illumina MiSeq platform as described previously [31]. The raw next-generation sequencing (NGS) data from this study can be found in the GenBank database under Project number PRJNA817054.

The Illumina MiSeq Fastq reads obtained were analyzed using the Quantitative Insights into Microbial Ecology 2 (QIIME 2) pipeline (version 2020.2; https://view.qiime2.org/ 4 April 2022) with default parameters unless otherwise noted [32,33,34]. Reads were processed by the DADA2 program using the qiime dada2 denoise-single script for denoising raw fastq single-end sequences, dereplicating, and chimera filtering [35]. Open reference operational taxonomic unit (OTU) picking was performed using VSEARCH and the SILVA v138 reference databases at 97% identity [36], which provides a feature table containing the frequencies of each OTU or taxon per sample [37]. Singletons were discarded for taxonomy and statistical analyses.

Differences among bacterial communities were calculated using rarefaction curves of alpha-diversity indexes (Observed OTUS, Faith_pd, Shannon) and beta diversity (Bray Curtis, Jaccard, Unweighted and Weighted UniFrac distances) at the OTUs level. Alpha and beta diversity as well as alpha rarefaction curves were conducted rarefying all samples to the minimum number of reads found (7393 sequences) to assess differences in microbial diversity between the four phases of the study (https://github.com/qiime2/q2-diversity, accessed on 4 April 2022). Finally, taxonomic, and compositional analyses were conducted by using the plugins feature-classifier classify-consensus-vsearch [38], and taxa barplot (https://github.com/qiime2/q2-taxa, accessed on 4 April 2022). Venn diagrams were generated using the “Venn diagram” online tool (http://bioinformatics.psb.ugent.be/webtools/Venn/, accessed on 4 April 2022) to identify shared (core microbiome) or unique taxa according to the phase of the study. 

### 2.9. Statistical Analysis

Statistical significances between the bread types for each aspect evaluated in the sensory questionnaires were obtained using the non-parametric Mann–Whitney–Wilcoxon test.

The non-parametric Kruskal–Wallis and PERMANOVA tests with FDR correction were used to find the existence of significant (*p* < 0.05) differences between the different phases in alpha and beta diversity indexes, respectively [39]. The differences in the relative abundance of bacterial taxa between the different periods of dietary intervention were tested using the non-parametric Mann–Whitney U test with SPSS Statistics for Windows Version 25.0 (IBM Corp., Armonk, NY, USA). For that, only taxa that were present in at least 70% of the samples per phase were used. 

A non-supervised multivariate hierarchical clustering analysis, using Euclidean distance and the Ward clustering algorithm, and a supervised principal least square-discriminant analysis (PLS-DA) of all bacterial taxa from the different phases of the study were performed using MetaboAnalyst 5.0 (http://www.metaboanalyst.ca, accessed on 4 April 2022; [40]). For that, only taxa that were present in at least 70% of the samples per each phase were used.

## 3. Results

### 3.1. Questionnaires

The sensory questionnaires evaluated four different aspects of the bread types consumed in the A–D phases: appearance, aroma, texture, and flavor. As expected, the gluten-containing bread of choice consumed in phase A exhibited significantly higher values in all aspects tested compared to the gluten-free bread consumed in phase B. Similar results were obtained when comparing both types of gluten-containing bread (phase A versus phase D); only texture scores had no significant differences between these phases, while the rest presented higher scores in phase A.

The scores of all sensory aspects improved significantly when the Tritordeum bread was incorporated into the gluten-free diet (phase C vs. phase B; Figure 2). In addition, the Tritordeum bread (phase C) presented higher scores for appearance than the gluten-containing bread of choice (phase D), whereas no significant differences were found in aroma, flavor, and texture between these types of bread. In contrast, the Tritordeum bread (phase C) presented significantly lower scores for aroma and flavor than the normal gluten bread consumed in phase A while no significant differences were found in appearance and texture.

### 3.2. Proteomic Data Analysis from Wheat and Tritordeum White Flours

Protein extraction of white flour from wheat and Tritordeum breads was performed and digested with chymotrypsin or trypsin. Then, peptides from gluten and non-gluten proteins (NGPs) were identified by LC-MS/MS analysis, in which custom Phyton scripts searched against annotated peptides from wheat and barley species, as one Tritordeum parent is a wild barley species.

As shown in Figure 3, both enzymatic treatments were effective in recognizing peptides from gluten proteins and NGPs. However, the number of peptides identified in each type of protein varied between both treatments, being this number higher for gliadin proteins when chymotrypsin was used, and for glutenin proteins and NGPs when trypsin was used (Figure 3). In the case of gluten proteins, there were more unique peptides assigned to these proteins in white wheat flour than in Tritordeum flour (Figure 3), regardless of the digestion enzyme used. This difference was more noticeable for α-gliadins. The same occurred in the case of amylase/trypsin-inhibitors (ATIs) and avenins; however, Tritordeum had more unique peptides assigned for the other NGPs such as globulins, plant lipid transfer proteins (LTPs), and serpins (Figure 3).

The custom scripts were also used to search for CD-epitopes, the p31–43 peptide, monoclonal antibodies (moAb) recognition sites, and the IgE binding sites in all peptides identified by LC-MS/MS after protein digestion (Appendix A). 

All CD-epitopes used in this study were defined by Sollid et al. [10,41]. The p31–43 peptide was added as it was reported of inducing the innate immune response in CD patients [42]. The moAbs recognition sites searched are the most used antibodies in the analysis of gluten peptides and proteins: R5, G12, and A1. The ELISA R5 method is considered a Type I analysis method by the Codex Alimentarius (CODEX STAN 118-1979) for the determination of gluten in food and is the method recommended by the Working Group on Prolamin Analysis and Toxicity (WGPAT). Additionally, its recognition sites are well described [43]. Both the G12 and A1 moAbs were designed to recognize the 33-mer peptide of α-gliadins and have shown a wider specificity for prolamins that are toxic to CD patients [44,45,46]. Finally, the IgE binding sites have been previously related to diverse diseases such as wheat allergy, baker’s asthma, wheat-dependent exercise-induced anaphylaxis (WDEIA), and atopic dermatitis in patients sensitized with wheat [47,48].

In this scenario, we studied those sequences with a perfect match and one mismatch found at any position of each peptide sequence, considering one match (event) or more in the same peptide sequence, as well as overlapping events. The results are displayed in Table 1. 

We observed that white flour of wheat bread had a higher number of events for all searched sequence types, regardless of the digestion enzyme utilized (Table 1). When no mismatches were allowed, these differences were more noticeable in the case of CD-epitopes, G12, and A1 moAb recognition sites which were 3.5-, 5.5-, and 3.8-times higher in wheat flour when compared to Tritordeum flour after chymotrypsin digestion, respectively (Table 1). After trypsin digestion, these differences were still observed for the number of CD-epitopes, G12 and A1 moAb recognition sites which were 2.6-, 2-, and 1-times higher in wheat flour, respectively. When a mismatch was allowed, white bread wheat flour retained the highest number of events in all types of sequence (Table 1). Additionally, we reported that recognition sites of moAbs used in this study were also present in CD-epitopes and the 33-mer (Appendix A). Specifically, G12 moAb is able to recognize CD canonical epitopes and 33-mer peptides while R5 recognizes also the p31–43 peptide (Appendix A). All these data suggest that Tritordeum is less immunogenic than bread wheat.

### 3.3. Diversity of Analysis of the Gut Microbiota

Illumina MiSeq sequencing analysis resulted in a total of 2,306,399 good-quality sequences retained after removal of chimeras and unassigned reads. Three samples were lost, corresponding to a repetition of phase D of subject number 7, a repetition of phase C of subject number 9 and a repetition of phase A of subject number 10. Satisfactory Good’s coverage was obtained for all samples with a mean value > 99.8%.

We did not find any significant differences in bacterial diversity between the different phases of study for any of the alpha diversity indexes estimated (Observed OTUS, *p* = 0.643; Shannon, *p* = 0.920; Faith_PD, *p* = 0.547) (Figure 4). In the same way, we did not find any significant differences for Bray Curtis (*p* = 0.361) and Unweighted (*p* = 0.867) Unifrac beta diversity distances (Figure 5) among the phases of study.

Both the PCoA (Figure 5) and hierarchical clustering analysis showed a trend to group the four dietary phases of each patient together, indicating the maintenance of the global structure and composition of the intestinal microbiota of each patient after consumption of a gluten free diet as well as after consumption of tritordeum and wheat breads (Appendix A). PLS-DA of all bacterial taxa showed a trend to progressively separate each phase according to its order of consumption; with phases A and D being the most different between each other (Figure 6A). PLS-DA identified 15 bacterial taxa as the most important in rank projection (VIP scores > 2.5 at *p* < 0.05; Figure 6B).

### 3.4. Relative Abundance and Changes in the Gut Microbiota

In total, 5 phyla, 8 classes, 16 orders, 27 families, and 41 genera, were identified with a mean relative abundance equal to or greater than 0.1% in at least one phase and with a presence equal to or greater than 70% in all samples in at least one phase. 

We found significant differences between the phases A and B in the relative abundance of *Lachnospirales* order (*p* = 0.033), *Lachnospiraceae* family (*p* = 0.020), and *Coprococcus* genus (*p* = 0.023) that were significantly higher in phase A than B. On the other hand, *Barnesiellaceae* (*p* = 0.018), *Marinifilaceae* (*p* = 0.004), and *Tannerrellaceae* (*p* = 0.022) families and *Barnesiella* (*p* = 0.019), *Odoribacter* (*p* = 0.023), and *Parabacteroides* (*p* = 0.028) genera showed a higher relative abundance in phase B than in phase A (Figure 7). We also observed significantly higher relative abundances on *Proteobacteria* phylum (*p* = 0.039), *Burkholderiales* order (*p* = 0.027), *Sutterellaceae* family, and the UCG-003 genus (*p* = 0.014) between the phases B and C (Figure 7). Finally, we found a significant higher relative abundance in phase C than on phases A for the *Marinifilaceae* family (*p* = 0.020) and *Eubacterium_ventriosum* group genus (*p* = 0.031) (Figure 7).

A total of 122 genera were shared between the 4 study phases (core microbiome) of a total of 199 genera (61,3%) (Figure 8B). However, some unique taxa were identified in each phase (Figure 8A,B). More specifically, for phase A, a total of 158 genera were identified, which 10 of them being unique (*Acetitomaculum*, *Acidibacter*, *Allorhizobium-Neorhizobium-Pararhizobium-Rhizobium*, *Brevundimonas*, *Candidatus Saccharimonas*, *Flavobacterium*, *Kocuria*, *Paracoccus*, *Peptostreptococcus*, and *Pseudomonas*). For phase B, a total of 160 genera were identified, and 9 were unique to this phase (*Aggregatibacter*, *Anaeroplasma*, *Asinibacterium*, *Devosia*, env, OPS_17, *Hafnia-Obesumbacterium*, *Lactonifactor*, *Thermoanaerobacterium*, and *Thermus*). For phase C, a total of 150 and 7 unique genera (*Catenibacillus*, *Enterococcus*, *Gemella*, *Massilia*, *Meiothermus*, *Micrococcus*, V9D2013_group) were identified. Finally, for phase D, 160 genera were identified with 10 being unique (*Acetanaerobacterium*, *Azospira*, *Cloacibacillus*, *Eubacterium*, *Exiguobacterium*, *Ezakiella*, GCA-900066575, *Lachnospiraceae*, *Papillibacter*, and *Sulfurimonas*) to this diet (Figure 8A).

In addition, we observed changes between phases. When challenging from phase A to B, we observed that both phases shared 141 genera, 17 genera were not detected, and 19 genera appeared with the consumption of GFD (Figure 8A). In the same way, when changing from phase B to C, we have detected that both phases shared 134 genera, 26 genera were not detected, and 16 genera appeared as new in phase C (Figure 8A). Finally, when we have compared phases C and D we detected that both phases shared 137 genera, 13 genera were not detected and 23 genera appeared as new in phase D (Figure 8A).

Finally, when analyzing only gluten-free diets (i.e., phases B, C, and D; Figure 8C) the bacterial core microbiome is composed by 128 genera, including six more genera than when considering the four phases. These genera are *Atopobium*, *Catabacter*, *Citrobacter*, *Coprobacter*, *Epulopiscium,* and *Fenollaria*.

## 4. Discussion

Wheat is the most used cereal for bread-making worldwide. This is mainly due to gluten proteins, which confer wheat dough with its unique viscoelastic characteristics [1]. However, gluten is also involved in a series of human disorders: gluten ataxia, dermatitis herpetiformis, NCGS, and CD, among others [2]. The existence and prevalence of these diseases prompted the development of gluten-free foods. An increasing fraction of healthy individuals currently believes that adhering to a GFD may have certain health benefits and, therefore, attempt to avoid gluten in their diet [49]. However, GFDs are frequently highly processed, expensive, and have also been reported to have detrimental effects on gut microbiota [21]. They may also increase fat and sugar intake and lack nutritional value compared to gluten-containing diets [50]. Tritordeum is a gluten-containing alternative cereal for bread making. This cereal has around half of the total gluten content of bread wheat and a different composition but with outstanding breadmaking quality [13]. Moreover, preliminary results suggest that tritordeum could be an option for a set of NCWS patients [28]. Furthermore, a recent pilot study also concluded that tritordeum foods could significantly reduce certain IBS symptoms by an overall improvement of the gastrointestinal barrier [29]. 

In this study, we reported the results of a four-phase study with healthy volunteers to determine changes in gut microbiota after consumption of regular gluten bread (phase A), gluten-free bread (phase B), a given tritordeum bread (phase C) and a given gluten-containing bread (phase D). Sensory questionnaires of the different bread types were also included and analyzed. Additionally, we characterized immunogenic epitopes, moAb recognition sites, and IgE binding sites of tritordeum flour and compared its composition with wheat flour.

The present study is a continuation of that by Vaquero et al. [13]. Their four-phase study also addressed the overall acceptability of tritordeum bread and compared it to the other three types of bread included in their study: gluten bread of choice, gluten-free bread, and a given gluten-containing bread. They observed a similar overall acceptance between tritordeum bread and the wheat bread of choice consumed by the healthy volunteers of the study. In the present study, we found comparable results; tritordeum bread had a statistically identical appearance and texture to wheat bread, but slightly lower flavor and aroma scores. Additionally, all sensorial aspects scored significantly higher in tritordeum bread when compared to gluten-free bread. This finding is also supported by Vaquero et al. [13] in which study participants also preferred tritordeum bread over gluten-free bread. Similar results were reported by Sánchez-León et al. [28], in which a set of NCWS patients rated tritordeum bread higher than the gluten-free bread they usually consumed. 

Vaquero et al. [13] stated that, compared to wheat bread, tritordeum breads had 49% less gluten and a 59 and 77% reduction in γ- and α-gliadin epitopes, respectively. Furthermore, tritordeum bread reported a high reduction in CD-epitopes. In this study, we further analyzed the content of immunogenic gliadin epitopes in tritordeum bread and wheat bread as we used an actualized database, which included newly revised epitopes [10]. Here, we reinforce the results obtained in previous studies; tritordeum flour presented a high reduction for CD-epitopes, but also for the recognition sites of G12, A1, and R5 moAbs and IgE binding sites. These findings are consistent with the reduction observed in the content of gluten immunogenic peptides (GIP) measured in the stool of healthy individuals after tritordeum bread compared to wheat bread [13]. All these results reinforce the idea that tritordeum has a lower immunogenic capacity than bread wheat. 

Regarding the results obtained from the analysis of the gut microbiota of these subjects in the four study phases, we observed that the global microbial profile remained relatively stable after a GFD and after the introduction of tritordeum bread and wheat bread in healthy individuals compared to a normal diet. In this sense, a core microbiome composed of 122 bacterial genera was detected. 

In addition, we did not find significant differences in either alpha or beta diversity indexes, which means that the consumption of normal bread, gluten-free bread, tritordeum bread, or wheat bread does not induce changes in alpha diversity of the microbial community of these subjects, at least during the 7 days duration of each phase in this study. Furthermore, beta diversity analysis also indicated a grouping of the different samples obtained in the four phases for the same subject. All these findings further reinforce the maintenance of the microbial community structure of these patients after the consumption of the four types of bread evaluated in this study. These results agree with Caio et al. [51], who observed a relatively stable microbial profile in healthy individuals with GFD.

However, although we have not found significant differences in global alpha and beta diversity indexes, some differences in the relative abundance of some specific taxa were observed among the four study phases, especially between phase A where subjects consumed a normal gluten-containing diet and phase B where subjects consumed a strict GFD. In this sense, we have observed that the genera *Parabacteroides*, *Barnesiella*, and *Odoribacter* increased their relative abundance after consumption of a GFD compared to a normal diet. These three bacterial genera are known for their function as producers of short chain fatty acids (SCFA) and in the degradation of intestinal mucus; therefore, they are bacterial genera involved in the maintenance of intestinal mucosa homeostasis [52,53,54,55]. On the contrary, the genus *Coprococcus* decreases significantly in the strict GFD phase compared to the normal diet, or phase A, and is also known as a butyrate producer [56]. These results show that gluten withdrawal produces shifts in the bacterial genera known to metabolize carbohydrates and starch as energy substrates, which has also been described by Caio et al. [51]. However, these differences were not observed after the introduction of tritordeum and wheat breads.

Interestingly, PLS-DA analysis showed that phase A and phase D were the most distant phases among the four of the study, showing the highest differences on the important taxa discriminating the phases, while phase B and phase C overlap. These results suggest a progressive change in the structure and composition of the intestinal microbiota from phase A to phase D. In this sense, the genera *Lachnospira*, *Roseburia*, *Lachnospiraceae-NK4A136_group*, and *Clostridia UCG-014* have a higher abundance in phase A and a lower abundance in phase D. On the contrary, *Butyricimonas*, *Turicibacter*, *Flavonifractor*, *UBA1819*, and *[Ruminococcus]_torques_group* have a higher abundance in phase D and a lower abundance in phase A. We have also observed that *Lachnospiraceae-UCG-003*, *[Eubacterium]_siraeum_group* and *Butyrivibrio* are less abundant in GFDs compared to a normal diet or phase A, especially in strict GFD or phase B. Additionally, members of the *Lachnospiraceae family*, such as *Lachnospira*, *Roseburia*, *Lachnospiraceae-NK4A136_group* decreased with a strict GFD and *Lachnospiraceae-UCG-003* and *Butyrivibrio* also decreased with phase D where wheat bread is incorporated into the GFD, as compared with a normal diet with gluten. All these bacterial taxa are considered the main producers of SCFA and therefore are related to intestinal health [57] and the observed changes are probably a consequence of the different starch composition of a GFD with a normal diet, as proposed by Bonder et al. [58]. However, we did not observe relevant changes with the introduction of tritordeum bread in phase C compared to a normal or GFD diet. According to these findings, tritordeum bread consumption could be considered for healthy adults without wheat-related diseases who wish to decrease their gluten intake without altering their gut microbiota. Unfortunately, other long-term dietary interventions in larger populations are needed to better assess the addition of tritordeum to a GFD.

## 5. Conclusions

In this work, we further characterized the nature of tritordeum flour peptides. When compared to bread wheat, tritordeum appeared to be less immunogenic as its peptides were assigned to less gluten proteins and presented fewer CD-related epitopes, monoclonal antibody recognition sites, and IgE binding sites. The 4-phase dietary intervention with healthy adults also revealed that subjects preferred tritordeum bread over gluten-free bread and found organoleptic properties between tritordeum and gluten-containing bread to be similar. Our microbiota analysis results reflect that, in general, the intestinal microbiota remains stable in healthy individuals after the consumption of a strict GFD, although involves some minor changes, especially altering the SCFA-producing species, in particular those involved specifically in carbohydrate and starch metabolism, which is in agreement with other studies [50,56,57]. However, short-term consumption of tritordeum bread in a GFD does not induce significant changes in the diversity or community composition of the intestinal microbiota in healthy individuals. These are important findings since it seems that tritordeum bread could be considered an alternative for those healthy adults without wheat-related pathologies who want to reduce their gluten consumption in the diet without harming their gut health. However, more studies are required to evaluate the introduction of tritordeum to a GFD for a longer period and in a larger population, both of healthy individuals and patients with any of the gluten-related diseases.

## Figures and Tables

**Figure 1 foods-11-01439-f001:**
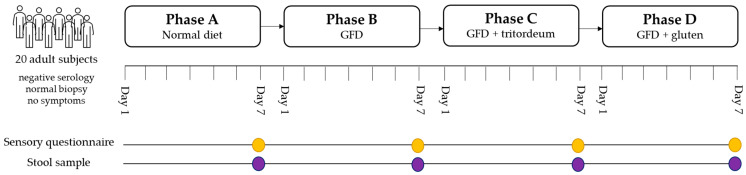
Study design. GFD: gluten-free diet.

**Figure 2 foods-11-01439-f002:**
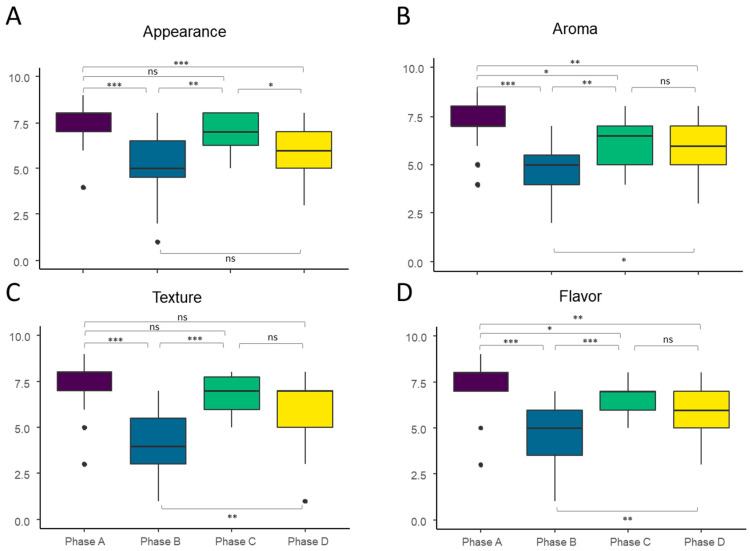
(**A**) Appearance, (**B**) aroma, (**C**) texture, and (**D**) flavor of dietary phases in the study. Phase A: normal diet and bread of choice with gluten, phase B: gluten-free diet and gluten-free bread, phase C: gluten-free diet and tritordeum bread, phase D: gluten-free diet and wheat bread. Statistical significances were obtained by Mann–Whitney–Wilcoxon test. * *p* < 0.05; ** *p* < 0.01; *** *p* < 0.001; ns, not significant.

**Figure 3 foods-11-01439-f003:**
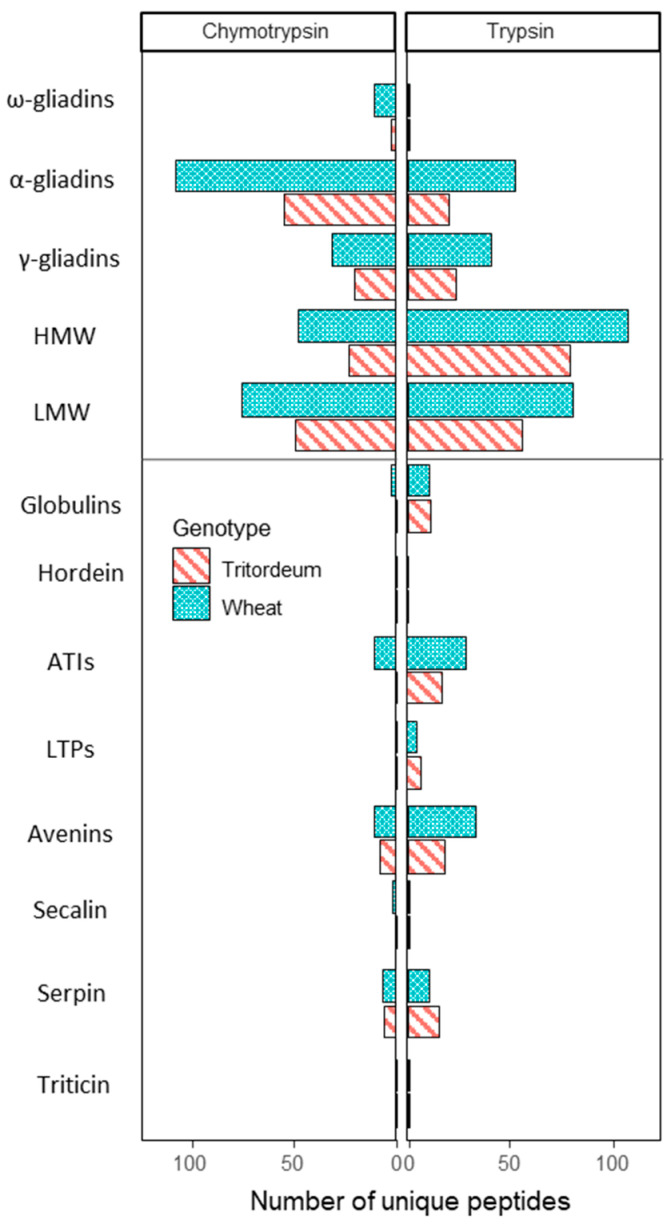
Number of unique peptides of grain protein components of white flour of bread wheat and tritordeum. The samples were digested by two different digestion enzymes. We considered peptides annotated as proteins of *Triticum aestivum*, *Triticum turgidum*, and *Hordeum* spp., with pep expect < 0.05, and from proteins supported for more than one peptide. Gluten proteins and NGPs were separated in the figure. NGPs: non-gluten proteins. HMW: high molecular weight glutenins. LMW: low molecular weight glutenins. ATIs: amylase/trypsin-inhibitors. LTPs: lipid transfer proteins.

**Figure 4 foods-11-01439-f004:**
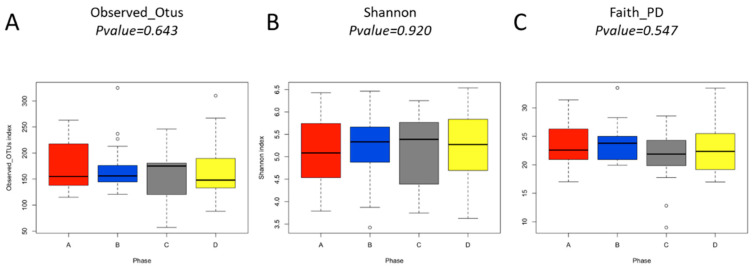
Results of the Alpha diversity analysis. (**A**) Observed Otus, (**B**) Shannon, and (**C**) Faith PD alpha diversity indexes for each phase of the study at a depth of 7393 sequences per sample. The *p*-value for each alpha diversity index was estimated using a Kruskal–Wallis test.

**Figure 5 foods-11-01439-f005:**
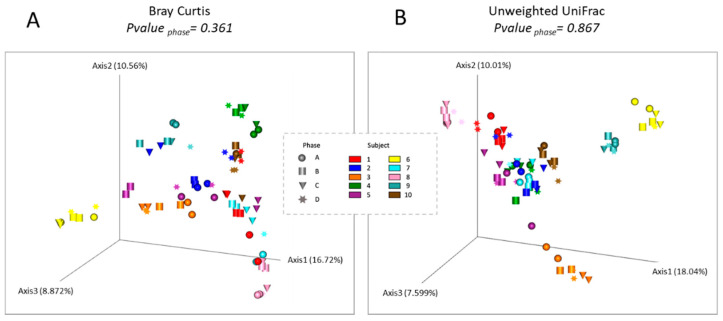
Principal coordinate analysis (PCoA) of the (**A**) Bray Curtis and (**B**) Unweighted UniFrac beta diversity distances of the four phases of the study. The *p*-value was estimated using a Permanova test. Each phase is represented by different shapes and each subject in a different color.

**Figure 6 foods-11-01439-f006:**
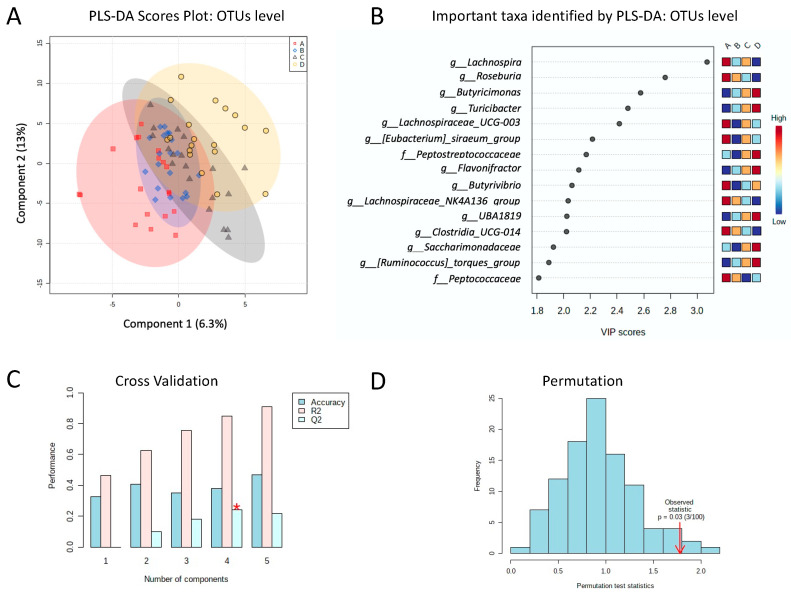
(**A**) Partial least square-discriminant analysis (PLS-DA) two-dimensional (2D) scores plot of the bacterial taxa of each phase of study at OTUS level. The model was established using two principal components; explained variance is in parentheses. (**B**) Loading importance of bacterial taxa in the first PLS-DA component at OTUS level. Colored boxes indicate relative concentrations of the corresponding bacterial taxa in each diet. (**C**) Cross Validation: PLS-DA classification using different number of components based on the accuracy (blue), R2 (pink), Q2 (light blue) by 10-fold CV method. The red star indicates the best classifier. (**D**) Permutation: PLS-DA model validation by permutation tests based on separation distance. The *p* value based on permutation is *p* = 0.03 (3/100). VIP: Variable Importance in Projection.

**Figure 7 foods-11-01439-f007:**
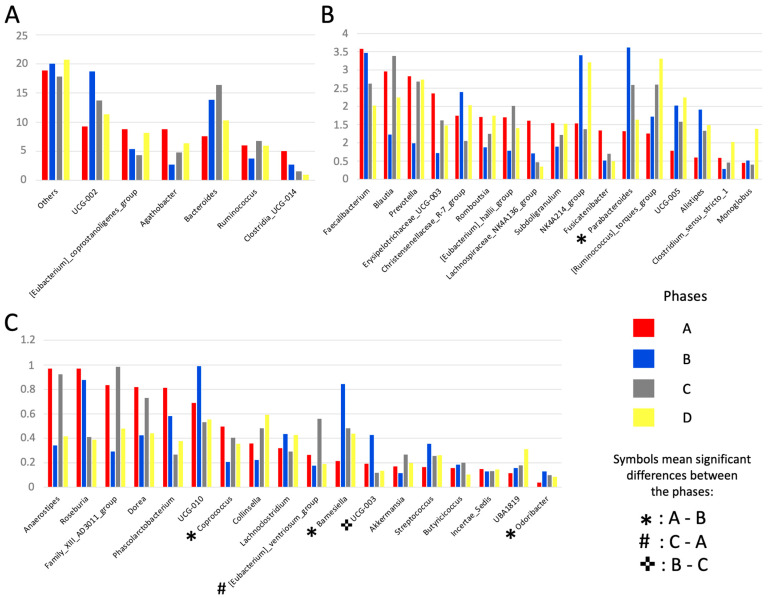
Bar plots of the relative abundance of each taxonomic group in percentage within each phase at genera level. (**A**) Taxa with a relative abundance ≥ 5% in at least one phase; (**B**) Taxa with a relative abundance ≥ 1% and <4% in at least one phase; (**C**) Taxa with a relative abundance < 1%. Others corresponds to the total sum of unassigned taxa. Symbols indicate significantly different taxa (*p*-value ≤ 0.05) by non-parametric Kruskal–Wallis test.

**Figure 8 foods-11-01439-f008:**
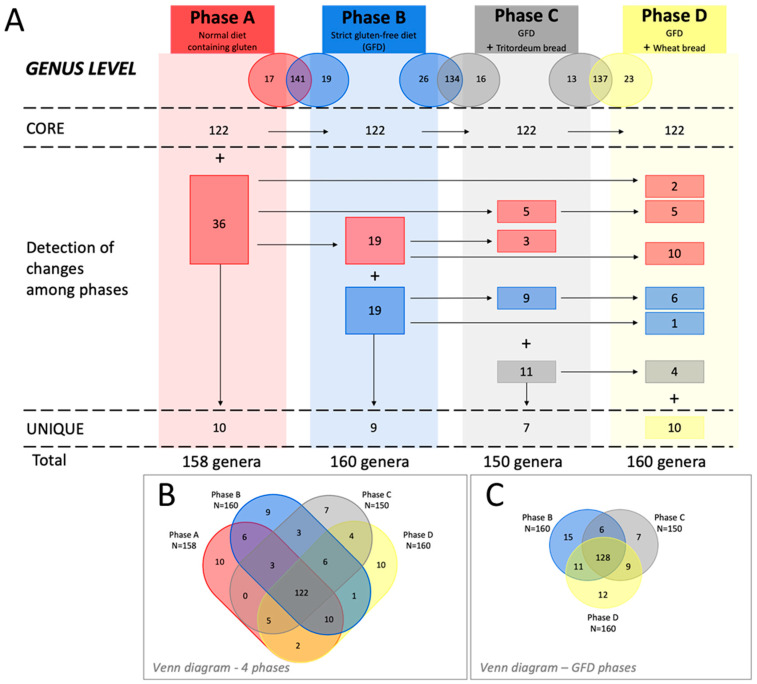
(**A**) Characterization of the temporal dynamics of the human gut microbiome throughout the four study phases at the genus level. (**B**) Venn diagram of the four phases of study. (**C**) Venn diagram of the three phases with a GFD. Each phase is represented with a color: red for phase A, blue for phase B, gray for phase C and yellow for phase D. The numbers that are represented in colored squares mean the number of genera that has been detected for the first time in the phase represented with the same color and the horizontal arrows represent the genera that are maintained from one phase to another.

**Table 1 foods-11-01439-t001:** Number of events with perfect match or one mismatch of CD immunogenic epitopes, recognition sites of monoclonal antibodies, and IgE binding sites in unique peptides ^1^ of different types of flour from wheat and Tritordeum by two digestion enzymes.

	Number of Events			
	Chymotrypsin		Trypsin	
Perfect Match	Wheat	Tritordeum	Wheat	Tritordeum
**CD epitopes**	49	14	64	24
**p31–43**	6	2	0	0
**G12**	22	4	3	1
**A1**	31	8	1	1
**R5**	27	8	8	2
**IgE recognition sites**	61	33	195	116
**1 mismatch**				
**CD epitopes**	47	15	73	33
**p31–43**	9	5	0	0
**G12**	112	31	52	22
**A1**	93	26	16	9
**R5**	156	64	71	31
**IgE recognition sites**	377	176	536	280

^1^ Unique peptides from proteins annotated for *Triticum aestivum*, *Triticum turgidum*, or *Hordeum* spp., with Pep_expect < 0.05 and from proteins supported for more than one peptide.

## Data Availability

The data presented in this study are available on request from the corresponding author.

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
