# Peer review of "Consumption of Tritordeum Bread Reduces Immunogenic Gluten Intake without Altering the Gut Microbiota"

_foods, 2022, doi:10.3390/foods11101439_

Round 1

Reviewer 1 Report

Interesting topic, there are minor problems in the text, eg the text usually cites the authors in one way (numbers, names), it is necessary to unify lines 181-182,298,450,459-460,463, 486 and 501. Also expressions of evidence with a value less than P = 0.01 given on line 375-385 is irrelevant, no corresponding conclusions and repeatability can be drawn from these values. Figure 1 is interesting but illegible. In the attached Sumpplementary (Table S2 and Table S1), modify the table to make sense.

Author Response

Reviewer 1: Interesting topic, there are minor problems in the text, eg the text usually cites the authors in one way (numbers, names), it is necessary to unify lines 181-182,298,450,459-460,463, 486 and 501.

Response to the reviewer: All cites have been corrected as suggested.

Reviewer 1: Also expressions of evidence with a value less than P = 0.01 given on line 375-385 is irrelevant, no corresponding conclusions and repeatability can be drawn from these values.

Response to the reviewer: In our study, we have used a p-value ≤ 0.05 to consider statistically significant differences. This probability value is the most common threshold value used to test for significant differences. So, all the bacterial taxa indicated in the text were significantly different. Maybe the reviewer considers that we need to select a smaller value?

Reviewer 1: Figure 1 is interesting but illegible.

Response to the reviewer: We inserted Figure 1 again in the text to solve this problem.

Reviewer 1: In the attached Supplementary (Table S2 and Table S1), modify the table to make sense.

Response to the reviewer: Table S1 includes all celiac disease (CD) and p31-43 epitopes, monoclonal antibodies recognition sites, and IgE binding sites used in this study, all associated with wheat-related pathologies. The table also includes the references in which they were previously described for further information. Table S2 describes the presence of all moAb recognition sites included in this study in CD canonical epitopes, p31-43 peptide, and the 33-mer peptide. Sequences found in any of these epitopes are shown in bold letters. Additionally, this table also includes the amount of these epitopes found after enzyme digestion (chymotrypsin and trypsin) in Tritordeum and wheat. The titles of both tables were modified to include more information.

“Table S1. Epitope names and sequences of CD-epitopes, p31-43, moAb recognition sites and IgE binding sites. References in which each one is described are also listed in the table. moAb: monoclonal antibody.”

“Table S2. Presence of each moAb recognition site in wheat and Tritordeum after chymotrypsin and trypsin digestion (left) and CD canonical epitopes, p31-43 peptide and the 33-mer peptide (right). Sequences found in any of these latter epitopes are shown in bold letters. moAb: monoclonal antibody.”

Reviewer 2 Report

The manuscript by Haro et al. "Consumption of Tritordeum bread reduces immunogenic gluten intake without altering the gut microbiotaLines 66-67: the sentence is unclear -please make proper amendments" aimed to characterize the immunogenic epitopes of tritordeum bread in a 4-phase study with healthy adults for preferences of bread and alterations in the gut microbiota after consuming wheat bread, gluten-free bread, and tritordeum bread. Tritordeum bread presented fewer peptides related to gluten proteins, CD-epitopes and IgE binding sites than bread wheat and was rated higher than gluten-free bread by study participants. Gut microbiota analysis revealed that the adherence to a strict GFD involves some minor changes, especially altering the species producing short-chain fatty acids. However, the short-term consumption of tritordeum bread did not induce significant changes in the diversity or community composition of the intestinal microbiota in healthy individuals. Therefore, they suggest that tritordeum bread could be an alternative for those who want to reduce their gluten consumption without harming their gut health.

Minor changes

Abstract and conclusion: it should be clearly specified to which groups you are referring that the tritordeum bread can be safely used as an alternative. and what do you mean with the short term consumption? 

Discussion: please add study limitations and strengths. the longer duration of consumption for future research.

LInes 70-78: it is appropriate to expand and make more specific this part. Probably the following reference can be of help doi: 10.1093/cdn/nzaa176

Author Response

Reviewer 2: Abstract and conclusion: it should be clearly specified to which groups you are referring that the tritordeum bread can be safely used as an alternative. and what do you mean with the short-term consumption? 

Response to the reviewer:

We very much appreciate the reviewer's comments. We rephrased specific sentences to specify the group of people that can safely consume Tritordeum. The last sentence of the abstract now reads like this: 'Therefore, our research suggests that tritordeum bread could be an alternative for healthy individuals without wheat-related pathologies who want to reduce their gluten consumption without harming their gut health.' We did the same in the conclusions: 'These are important findings since it seems that tritordeum bread could be considered an alternative for those healthy adults without wheat-related pathologies who want to reduce their gluten consumption in the diet without harming their gut health.'

We can only assume that Tritordeum does not affect gut microbiota in healthy adults after a short-term consumption lasting a week; this was the dietary intervention period stated in the Methods section. Further studies would be needed to maintain the same results with more extended consumption periods.

Reviewer 2: Discussion: please add study limitations and strengths. the longer duration of consumption for future research.

Response to the reviewer: Thank you for your comment. We added a few sentences at the end of the Discussion section indicating the limitations of our short-term study. They are also stated in the Conclusions.

Reviewer 2: Lines 70-78: it is appropriate to expand and make more specific this part. Probably the following reference can be of help doi: 10.1093/cdn/nzaa176

Response to the reviewer: Thank you very much for your suggestion. We added a few references from the review.